# MoMa: Efficient Early-Fusion Pre-training with Mixture of Modality-Aware Experts

## Abstract

We introduce MoMa, a novel modality-aware mixture-of-experts (MoE) architecture designed for pre-training mixed-modal, early-fusion language models. MoMa processes images and text in arbitrary sequences by dividing expert modules into modality-specific groups. These groups exclusively process designated tokens while employing learned routing within each group to maintain semantics-based adaptivity. Our empirical results reveal substantial pre-training efficiency gains through this modality-specific parameter allocation. Under a 1-trillion-token training budget, the MoMa 1.4B model, featuring 4 text experts and 4 image experts, achieves impressive FLOPs savings: $3.7\times$ overall, with $2.6\times$ for text and $5.2\times$ for image compared to a compute-equivalent dense baseline, measured by pre-training loss. This outperforms the standard expert-choice MoE with 8 mixed-modal experts, which achieves $3\times$ overall FLOPs savings ($3\times$ for text, $2.8\times$ for image). These results demonstrate MoMa's potential to significantly advance the efficiency of mixed-modal, early-fusion language model pre-training, paving the way for more resource-efficient and capable multimodal AI systems.

## 1 Introduction

Auto-regressive mixed-modal foundation models have shown significant promise in applications requiring the processing of mixed-modal inputs and the generation of mixed-modal outputs (Gemini et al., 2023; 2024; OpenAI et al., 2024; Lu et al., 2023; Chameleon Team, 2024). These models have demonstrated remarkable capabilities in tasks ranging from visual question answering to multimodal content generation, pushing the boundaries of AI's ability to understand and interact with our inherently multimodal world.

While a popular architecture design for mixed-modal foundation models involves fusing modality-specific encoders or decoders (Gemini et al., 2023; 2024; Lu et al., 2023; OpenAI et al., 2024), this approach can limit the model's ability to integrate information across modalities and generate content with interleaved modalities. To address this limitation, `Chameleon` (Chameleon Team, 2024) recently introduced a single transformer architecture with a next-token prediction objective to model mixed-modal sequences consisting of discrete image and text tokens, allowing for seamless reasoning and generation across modalities. However, scaling such mixed-modal early-fusion foundation models to greater capacities presents significant computational challenges.

To address these challenges, we investigate the application of routed sparse architectures (Lepikhin et al., 2020; Fedus et al., 2022; Clark et al., 2022; Jiang et al., 2024; Raposo et al., 2024). These architectures have previously shown effectiveness in scaling language and vision-specific foundation models, as well as multimodal contrastive learning (Mustafa et al., 2022). However, their application to mixed-modal early-fusion models presents unique opportunities and challenges.

The insight driving our approach is the inherent heterogeneity of modalities: text and image tokens possess distinct information densities and redundancy patterns (Liang et al., 2023). While we integrate these tokens into a unified early-fusion architecture, we propose further optimizing this framework by incorporating modality-specific modules. This concept, which we term *modality-aware sparsity*, enables models to better capture features specific to each modality while maintaining strong cross-modality integration through partial parameter sharing and attention mechanisms. Previous work such as VLMo (Bao et al., 2022), BEiT-3 (Wang et al., 2022b) and VL-MoE (Shen et al.,

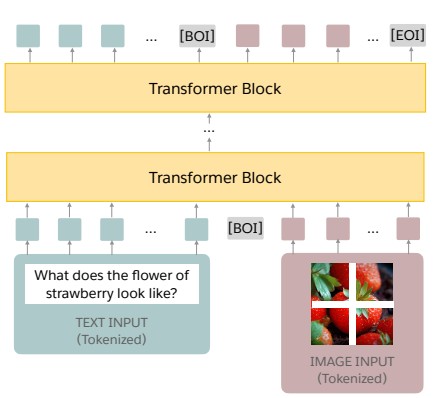
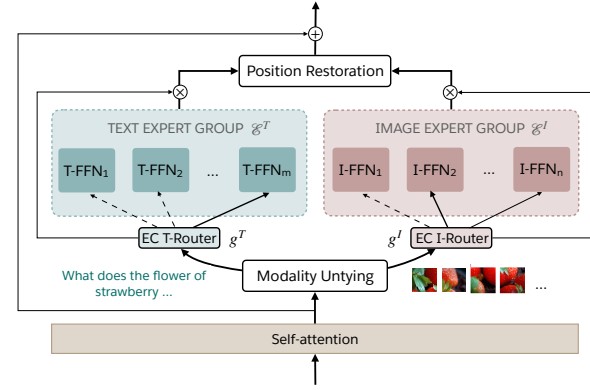

(a) Early-fusion mixed-modal LLM architecture.     (b) Mixture of modality-aware experts (MoMa) transformer block.

Figure 1: Overview of our proposed architecture. (a) The model processes interleaved text and image data as a sequence of discrete tokens using a transformer. (b) Illustration of the mixture of modality-aware experts (MoMa) transformer block, where layer norms and the residual connection of attention layer are omitted for readability.

2023) have applied modality-aware experts to train vision-language encoders and masked language models. We extend this approach to the mixed-modal, early-fusion language modeling setting.

Specifically, we adopt `Chameleon` as the base architecture and apply *mixture-of-experts* (MoE) (Lepikhin et al., 2020) where tokens are routed across a set of feed-forward blocks (experts) at each layer. Crucially, we divide these experts into modality-specific groups, with each group processing only tokens of its designated modality while learned routing is done within each group for semantics-based adaptivity. We dub this approach *mixture of modality-aware experts* (MoMa). We adopt expert-choice routing (Zhou et al., 2022) to ensure load balancing and maintain a static computation graph (Paszke et al., 2019), both important for high training throughput.

We conduct extensive FLOPs-controlled experiments comparing our proposed architecture with the dense baseline and multiple sparse variations. With a 1-trillion-token training budget, MoMa 1.4B with 4 text experts and 4 image experts achieves a substantial $3.7\times$ overall FLOPs savings (text: $2.6\times$, image: $5.2\times$) compared to the 1.4B isoFLOP dense baseline as measured by pre-training loss, while maintaining a relatively modest -17% throughput reduction. In contrast, the standard expert-choice MoE with 8 mixed-modal experts achieves $3\times$ FLOPs savings (text: $3\times$, image: $2.8\times$) under the same setting, with -9% throughput reduction. These results demonstrate MoMa's potential to significantly advance the efficiency of mixed-modal, early-fusion language model pre-training, paving the way for more resource-efficient multimodal AI systems.

## 2 MODEL

### 2.1 EARLY FUSION

Our model builds upon the early fusion architecture introduced by `Chameleon` (Chameleon Team, 2024), which represents images and text as a series of discrete tokens within a unified transformer. The core of `Chameleon` is a transformer-based model that applies self-attention mechanisms over the combined sequence of image and text tokens. This allows the model to capture complex relationships between and within modalities. The model is trained using a next-token prediction objective, learning to generate both text and image tokens autoregressively.

In `Chameleon`, images are tokenized using a learned image tokenizer that encodes a $512 \times 512$ image into 1024 discrete tokens from a codebook of size 8192. Text is tokenized using a BPE tokenizer with a vocabulary size of 65,536, which includes the 8192 image codebook tokens. This unified tokenization scheme enables the model to process arbitrary sequences of interleaved image and text tokens.

By adopting this early fusion approach, our model inherits several key advantages:

1. *Unified representation:* The model learns a shared representation space for images and text, facilitating cross-modal reasoning and generation.

2. *Flexibility:* The architecture can handle arbitrary sequences of images and text, enabling diverse multimodal tasks such as image captioning, visual dialogue, and mixed-modal document generation.

3. *Scalability:* The token-based approach allows for uniform processing of both text and image data, enabling efficient scaling to larger model sizes and diverse datasets. This is evidenced by `Chameleon`'s successful training on approximately 10 trillion mixed-modal tokens.

4. *End-to-end learning:* The entire auto-regressive model, is trained end-to-end, allowing for joint optimization of the representation and task-specific performance.

Building on this foundation (Figure 1a), our work introduces modality-aware sparsity techniques to further enhance the efficiency and performance of early fusion models, as detailed in the following sections. These techniques aim to address the computational challenges of scaling early fusion models while maintaining their powerful cross-modal reasoning capabilities.

## 2.2 MIXTURE OF MODALITY-AWARE EXPERTS

We propose a width scaling approach that incorporates modality-aware block sparsity in the feedforward module, extending the standard mixture-of-experts (MoE) architecture (Lepikhin et al., 2020; Fedus et al., 2022; Wang et al., 2022a). The insight driving this approach is that tokens of different modalities have distinct characteristics and information densities. By creating separate expert groups for each modality, we allow the model to develop specialized processing pathways while maintaining the ability to integrate information across modalities.

We describe the key components in the mixture of modality-aware experts (MoMa) formulation below (Figure 1b).

**Modality-Specific Expert Groups.** We divide the experts in each MoE layer into distinct groups, each specialized in processing tokens from a specific modality $\boldsymbol{E} = \{E_1^{\boldsymbol{T}} \ldots, E_m^{\boldsymbol{T}}, E_1^{\boldsymbol{I}} \ldots, E_n^{\boldsymbol{T}}\}$: one group for processing text tokens and another for image tokens[1]. This separation allows each group to specialize in features relevant to its respective modality.

By implementing modality-aware block sparsity, we aim to achieve several benefits:

- *Improved efficiency:* by routing tokens to modality-specific experts, we reduce the computational overhead of processing tokens with experts not specialized for their modality.

- *Enhanced specialization:* modality-specific expert groups can develop more refined features relevant to their respective modalities.

- *Maintained Cross-Modal Integration:* despite the separation into modality-specific groups, the model still can integrate information across modalities through the shared self-attention mechanisms in non-MoE layers.

**Hierarchical Routing.** We adopt a token-based routing mechanism (Lepikhin et al., 2020; Fedus et al., 2022; Jiang et al., 2024). For an input token $x$, our routing mechanism operates in two stages.

1. *Modality-aware routing:* tokens are first routed to their corresponding modality-specific expert group based on their modality (text $\boldsymbol{T}$ or image $\boldsymbol{I}$).

2. *Intra-modality routing:* within each modality-specific expert group $\boldsymbol{E}^M$, tokens are then routed to specific experts using a learned routing function. Specifically, we use a projection matrix $W_g^{\boldsymbol{M}} \in \mathbb{R}^{d \times |\boldsymbol{E}^M|}, d =$ transformer hidden dimension, to compute the token-to-expert affinity scores, following other established MoE formulations (Lepikhin et al., 2020; Fedus et al., 2022; Jiang et al., 2024).

---

[1]While we focus on text and image modalities in this work, our formulation is designed to be easily extensible to accommodate an arbitrary number of modalities.

**Expert choice.** Within each modality group, we implemented expert-choice (EC) routing (Zhou et al., 2022), where each expert has a fixed bucket size and processes the top-$k_e$ tokens in a batch. We set $k_e = b^{\boldsymbol{M}} \cdot c_e$, where $c_e = \frac{1}{|\boldsymbol{E^M}|}$ is a capacity factor and $b^{\boldsymbol{M}}$ is the total number of tokens of modality $\boldsymbol{M}$ in a batch. As a result, each token can be routed to a variable number of experts.

EC routing ensures balanced expert utilization during training and eliminates needing a separate load-balancing loss term. Maintaining a single loss term promotes more stable optimization and a more straightforward training process. However, EC routing compromises causality in auto-regressive language modeling, as each expert selects the top tokens to process in a batch by comparing token scores across all tokens. We use a combination of two techniques to address this issue and enable expert-choice training for auto-regressive LMs.

1. We use the sigmoid function as the non-linearity in the router scoring function, enabling independent calculation of token-to-expert affinity scores for each token.

2. We introduce auxiliary routers, inspired by Raposo et al. (2024), which predict the likelihood of an expert selecting a token solely based on its hidden state representation. These routers are trained after the main model training is completed and employed during inference to ensure causality. We discuss the details of the auxiliary routers in Appendix A.2.

In summary, the MoMa module for an input token $x$ can be formally defined as:

$$y = \begin{cases} \sum_{j=1}^{m} g^{\boldsymbol{T}}(x)_j \cdot \text{FFN}_{\text{SwiGLU}_j}^{\boldsymbol{T}}(x) & \text{if } x \in \boldsymbol{T} \\ \sum_{j=1}^{n} g^{\boldsymbol{I}}(x)_j \cdot \text{FFN}_{\text{SwiGLU}_j}^{\boldsymbol{I}}(x) & \text{if } x \in \boldsymbol{I} \end{cases} \tag{1}$$

$$g^{\boldsymbol{M}}(x)_j = \begin{cases} \sigma(x \cdot W_g^{\boldsymbol{M}})_j & \text{if } x \text{ is selected by } E_j \\ 0 & \text{otherwise} \end{cases}$$

We further apply residual connection and the Swin Transformer normalization (Liu et al., 2022b) post MoMa computation. Our experiments, detailed in later sections, demonstrate that MoMa significantly improves efficiency and performance compared to dense baselines and standard MoE architectures.

### 2.3 AUXILIARY ROUTERS

We cannot directly apply the expert-choice routing for MoE during inference time, as the top-$k$ token selection within a batch breaks causality. Inspired by Raposo et al. (2024), to ensure causality during inference, we introduce auxiliary routers, to predict the likelihood of a token being selected by an expert or a layer solely based on its hidden representation.

Formally, for each MoE layer, we introduce an auxiliary router

$$g_{\text{aux}}^{\boldsymbol{M}}(x) = \sigma(\text{SiLU}(x \cdot W_{a1}^{\boldsymbol{M}}) \cdot W_{a2}^{\boldsymbol{M}}), \tag{2}$$

where $W_{a1}^{\boldsymbol{M}} \in \mathbb{R}^{d \times (d/2)}$ and $W_{a2}^{\boldsymbol{M}} \in \mathbb{R}^{(d/2) \times |\boldsymbol{E^M}|}$.

We employ a two-stage training approach, where the main model and auxiliary routers are trained separately. First, we train the main model to convergence. Then, we train the auxiliary routers using binary cross-entropy loss, supervised by the ground-truth top-$k$ routing assignments computed over an entire batch. At inference time, the main routers are only used to generate the weight values, and tokens are selectively routed to an expert or layer based on thresholding on the auxiliary router score ($> 0.5$).

## 3 EFFICIENCY OPTIMIZATION

To facilitate the distributed training of mixture of modality-aware experts (MoMa), we employ Fully Sharded Data Parallel (FSDP) (Zhao et al., 2023). This approach presents unique efficiency challenges compared to vanilla MoEs. We discuss these challenges and our strategies to address them in this section.

## 3.1 LOAD BALANCING

Without constraints, load imbalance can occur in our system because the ratio of text to image tokens can vary significantly across different GPUs and iterations. Imbalances can create a cascading straggler effect, delaying weight prefetching for subsequent layers and gradient releases from previous layers. This effectively bounds the training latency by the maximum time required to process text and image experts across all GPUs within a batch.

To mitigate these issues, we developed a balanced data mix that aligns the text-to-image data ratio with the expert ratio on each GPU. This approach ensures load balancing in expectation. While alternative rebalancing algorithms, such as token redistribution at each FFN layer, are possible, they may introduce additional communication overhead.

## 3.2 EFFICIENT EXPERT EXECUTION

We explored several strategies to efficiently execute experts for different modalities. The first approach restricts to homogeneous experts across modalities and prohibits routing of text tokens to image experts and vice versa. This method allows processing of all tokens and modalities simultaneously, provided all experts share the same token count. Alternatively, we could enhance execution efficiency by employing block sparsity (Gale et al., 2023), which offers similar benefits to the first approach without requiring perfect expert token balance. Additionally, we considered running experts from different modalities sequentially when the number of modalities is limited. This approach allows better overlap of computation from previous modality experts with weight prefetching for subsequent modality experts, alleviating memory pressure. It also removes assumptions about expert load balance.

Given that our experiments involve a sufficiently large token count per GPU, hardware utilization is not a major concern even with multiple batched matrix multiplications. Thus, we find the sequential approach to be a clean and flexible choice for our experimental environment at the current scale.

## 4 EXPERIMENTS

### 4.1 SETUP

We use the same pre-training dataset and preprocessing as Chameleon Team (2024). To assess scaling performance, we train all models with over 1 trillion tokens. Unless specified otherwise, we employ a sequence length of 4096 tokens and a model parallel size of 1. Our training regimen includes a peak learning rate of $1e - 4$, a 4000-step warm-up period, and linear annealing of the learning rate to 1% of its peak value. For all MoE architectures, we implement MoE in every layer, setting each expert's training capacity $k_e$ to $\frac{b^M}{|E^M|}$ to maintain FLOPs per token comparable to the base dense model. To achieve FLOPs parity with the base dense model, we increase the total layer count while maintaining a constant hidden dimension. Table 1 provides detailed configurations for our dense and sparse models. Additional pre-training details are available in Appendix A.1.

For model comparison, we report training losses. Given that our 1-trillion-token training budget covers less than one epoch of our extensive pre-training data, we use training loss as a proxy for validation performance. Our use of expert-choice routing in both MoE and MoD modules introduces a caveat: the training loss calculation compromises causality, as token selection considers the top proportion from a batch, including future tokens. We address this in §4.5 by reporting test-time performances using auxiliary routers, demonstrating result generalization to validation settings and causal scenarios.

### 4.2 SCALING OF PERFORMANCE WITH COMPUTE

We present the scaling performance of various models across multiple compute levels, with FLOPs matched to three dense model sizes: 90M, 435M, and 1.4B parameters. We report two key metrics:

Table 1: Specifications of dense and sparse architectures used in our experiments. Architectures listed in the same row are compute-matched with the same amount of active FLOPs per token. *params*: total number of parameters; $l$: number of layers; $d$: transformer hidden dimension; $ffn$: feed-forward module hidden dimension; $h$: number of attention heads; $e$: number of experts; $i$: MoE layer interval; $c$: MoE expert capacity.

| | Dense | | | | MoE | | | |
|---|---|---|---|---|---|---|---|---|
| *params* | $l$ | $d$ | $ffn$ | $h$ | *params* | $e$ | $i$ | $c_e$ |
| 90M | 8 | 512 | 2048 | 8 | 210M | 8 | 1 | 0.125 |
| 435M | 24 | 1024 | 4096 | 16 | 1.9B | 8 | 1 | 0.125 |
| 1.4B | 24 | 2048 | 8192 | 16 | 7.1B | 8 | 1 | 0.125 |

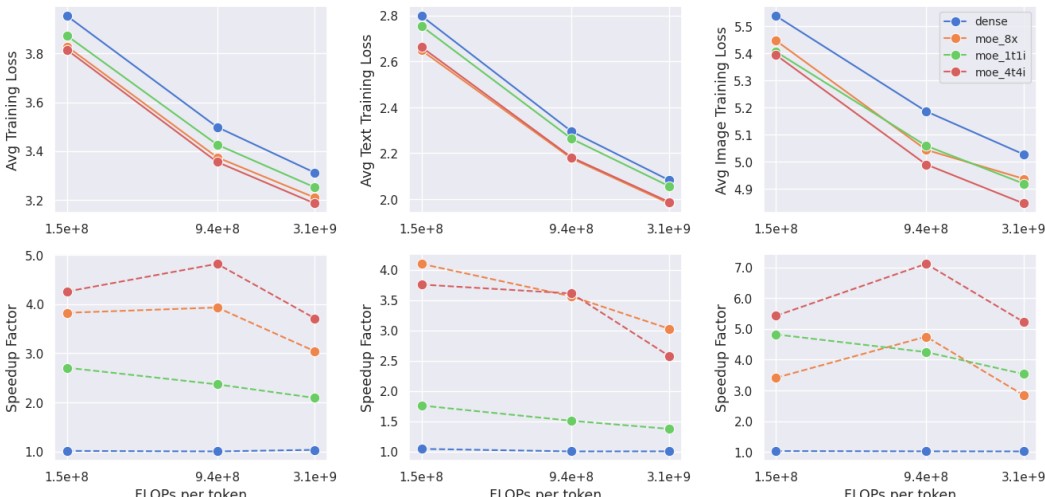

Figure 2: Scaling of performance with compute. We consider three dense model sizes – 90M, 435M and 1.4B parameters – and their isoFLOP sparse variations. We highlight three configurations of mixture-of-experts (MoE): 1) using 1 text expert and 1 image expert (moe_1t1i), 2) using 4 text experts and 4 image experts (moe_4t4i), and 3) using 8 mixed-modal experts (moe_8x) in addition to the dense baseline. All models were pre-trained and annealed to 1 trillion tokens. Controlled for FLOPs per token, architectures with modality-specific experts consistently outperforms the baselines, with particular large benefits shown on the image modality.

(1) training loss and (2) pre-training speed-up factor $\eta$, which indicates that a sparse model can match the pre-training loss of the isoFLOP dense baseline using only $1/\eta$ of total FLOPs[2].

**Modality untying.** Introducing modality-specific expert groups improves pre-training efficiency across different scales, with particularly large benefits for the image modality. As shown in Figure 2, the moe_1t1i configuration, using 1 image expert and 1 text expert, significantly outperforms the dense baseline. The image loss of moe_1t1i nearly matches that of the regular expert-choice MoE model using 8 mix-modal experts (moe_8x), whereas the text loss remains substantially higher, indicating parameter untying has a disproportionate impact on the image modality. Scaling the number of experts within each modality group further improve the model performance. The moe_4t4i configuration, using 4 text experts and 4 image experts, consistently outperforms moe_8x across different scales, with a substantial margin in image loss. However, this comparison reveals a slight regression in the text modality, suggesting that processing text modality across more experts may be beneficial.

---

[2]Our definition of $\eta$ is analogous to the speed-up factor proposed by Artetxe et al. (2021), but is defined in terms of pre-training loss whereas the original definition uses validation perplexity.

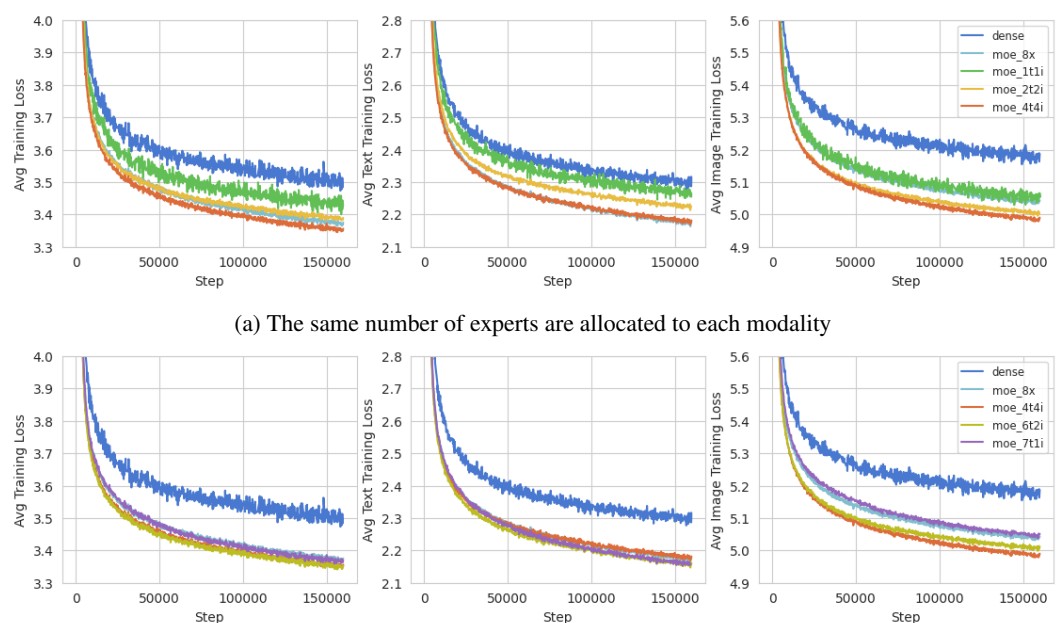

(a) The same number of experts are allocated to each modality

(b) Uneven number of experts are allocated to each modality

Figure 3: Training curves of 435M isoFLOP MoE architectures with increasing number experts and varied expert allocation across modalities.

## 4.3 SCALING NUMBER OF EXPERTS

We conducted further ablation experiments to investigate the impact of scaling the number of experts. We explored two scenarios: allocating an equal number of experts per modality (balanced expert allocation) and allocating different numbers of experts per modality. We included the dense model and expert-choice MoE with 8 mixed-modal experts (moe_8x) as baselines.

**Balanced expert allocation.** Figure 3a demonstrates that training loss consistently improves as the number of experts increases. Text and image losses exhibit distinct scaling patterns. While text loss improves steadily with each doubling of expert count, image loss shows diminishing returns as the number of experts increases from 2 to 4. This suggests that the intrinsic characteristics of each modality lead to different behaviors in sparse modeling. This finding aligns with our previous observation in §4.2, where untying modalities over 8 experts significantly enhanced image performance but not text performance, indicating that the text modality may benefit more from a larger number of experts.

**Imbalanced expert allocation.** Given the diminishing returns observed when allocating more than 2 experts to the image modality, we experimented with configurations allocating fewer experts to the image modality and more to the text modality. Figure 3b compares three configurations with the same total number of experts (8), varying the allocation between text and image modalities (moe_7t1i, moe_6t2i and moe_4t4i). We observed that allocating more experts to a modality generally enhances its performance. However, assigning more than 4 experts to the text modality also yields diminishing returns. The overall training losses of the three configurations converge to similar levels. Considering the better load balancing of balanced expert assignment with our pre-training data mixture, we selected moe_4t4i as our optimal MoE configuration, despite moe_6t2i having a slightly lower average loss. We leave the design of sparse architectures that can effectively leverage the intrinsic characteristics of different modalities to future work.

Table 2: Training throughput (wps/GPU) of various architectures compared with the same FLOPs as a 435M dense model on 256 A100 GPUs. Modality-conditioned sparsity offers a good quality-throughput tradeoff compared to the dense model, and exhibited scalability with more experts.

| Configuration | WPS | % diff |
|---|---|---|
| Dense | 31,970 | – |
| MoMa 8X | 28,990 | -9% |
| MoMa 1t1i | 30,196 | -6% |
| MoMa 4t4i | 26,424 | -17% |

## 4.4 THROUGHPUT ANALYSIS

Sparse models often struggle to fully achieve their theoretical efficiency advantages due to the additional computation complexity and data balancing challenges (§3). To quantify the impact of our proposals on training efficiency, we conducted a controlled experiment comparing the training throughput of various architectures. We compare architectures FLOPs-matched to the 435M dense baseline. We performed our experiments on 256 A100 GPUs with a sequence length of 4096 and a batch size of 6 per GPU and summarized the model throughput in Table 2.

Comparing expert-choice MoE (`moe_8x`) and dense models, introducing sparsity does incur overheads (-9%). The throughput loss likely comes from the need to compute routing decisions and synchronize gradients for all experts despite being FLOPs-equivalent. On the other hand, as discussed in §3, running experts sequentially by modality (`moe_1t1i`) does not suffer from large execution overheads, and we can attribute most of the loss (-6%) to computing token indices for each modality, which can be amortized by precomputing the indices and share with each transformer layer.

When combining modality conditioned feed-forward with learned routing (`moe_4t4i`), we observe a smooth degradation in throughput as the number of experts increases, incurring additional 11% overheads with 8 experts, which is on par with the throughput loss of 9% when transitioning from the dense model to MoE with 8 experts.

## 4.5 INFERENCE-TIME PERFORMANCE

We evaluate four models (1.4B: `dense`, `moe_1t1i`, `moe_8x`, `moe_4t4i`) on held-out language modeling data and downstream tasks. We report perplexity on subsets of OBELICS (Laurençon et al., 2023) and Shutterstock[3], which are held-out from our pre-training dataset. In addition, we report 0-shot performance on a set of commonsense reasoning of tasks commonly used for benchmarking pre-trained language models. We also selected several vision-language task datasets and report the perplexity of the ground truth output in these datasets for cross model comparison, where we format the examples in a zero-shot manner.[4]

- **Commonsense Reasoning:** We evaluated on a set of text-only tasks commonly used for benchmarking the commonsense reasoning capabilities of language models: `PIQA` (Bisk et al., 2020), `SIQA` (Sap et al., 2019), `HellaSwag` (Zellers et al., 2019), `WinoGrande` (Sakaguchi et al., 2021), `ARC-Easy` (Clark et al., 2018), `ARC-Challenge` (Clark et al., 2018), `OpenBookQA` (Mihaylov et al., 2018), and `BoolQ` (Clark et al., 2019). We score the prompt with each candidate's answer and compute accuracy using the candidate with the highest score.

- **Image Captioning:** We take the Karpathy test split of MS-COCO (Lin et al., 2014) and the Karpathy test split of Flickr30k (Plummer et al., 2015), and report text-to-image and image-to-text conditional perplexity using these two datasets.

- **Visual Question Answering:** We report the perplexity of ground truth answers on the test-dev split of VQA-v2 (Goyal et al., 2017).

---

[3] https://www.shuitterstock.com/

[4] We do not report the generation evaluation metrics for these tasks, as our experiment checkpoints are undertrained and have not achieved sufficient image understanding and generation capabilities to produce meaningful results on downstream tasks.

Table 3: Interleaved data modeling performance.

| Model | Image to text (PPL ↓) | | | Text to image (PPL ↓) | | Interleaved (PPL ↓) | | | | | |
| | COCO | Flickr | VQAV2 | COCO | Flickr | Obelics Overall | Obelics Text | Obelics Image | SSTK Overall | SSTK Text | SSTK Image |
|---|---|---|---|---|---|---|---|---|---|---|---|
| 1.4B Dense | 21.7 | 28.6 | 20.0 | 458.3 | 559.4 | 44.6 | 14.3 | 64.7 | 238.5 | 48.7 | 245.8 |
| 1.4B MoMa 1t1i | 21.9 | 29.3 | 19.8 | 416.7 | 508.6 | 41.5 | 13.8 | 59.3 | 208.1 | **7.6** | 221.5 |
| 1.4B MoE 8x | **18.9** | **24.2** | **18.9** | 426.5 | 518.5 | 40.4 | **12.5** | 59.2 | 218.8 | 33.7 | 226.6 |
| 1.4B MoMa 4t4i | 20.9 | 26.6 | 19.8 | **392.5** | **479.6** | **39.0** | 13.0 | **55.8** | **194.7** | 10.5 | **205.7** |

Table 4: Text-to-text commonsense reasoning task performance.

| Model | BoolQ | PiQA | SiQA | Winogrande1.1 | OBQA | Hellaswag | Arc-E | Arc-C | Avg |
|---|---|---|---|---|---|---|---|---|---|
| 1.4B Dense | 61.8 | 72.1 | 44.3 | 54.2 | 42.0 | 51.0 | 59.8 | 34.3 | 52.4 |
| 1.4B MoMa 1t1i | 61.5 | 71.3 | 44.4 | 54.7 | 46.0 | 52.8 | 61.1 | 34.9 | 53.3 |
| 1.4B MoE 8x | 62.2 | **73.5** | 44.4 | **58.3** | 42.2 | **58.2** | **62.6** | **36.9** | **54.8** |
| 1.4B MoMa 4t4i | **62.3** | 72.5 | **44.6** | 55.9 | **47.0** | 55.2 | 60.7 | 35.7 | 54.2 |

For all sparse models, we perform the second-stage training of auxiliary routers and use the auxiliary routers for causal inference. The details of auxiliary router training can be found in Appendix A.2.

**Interleaved data modeling.** According to Table 3, the relative performance of the dense model and various MoE configurations is consistent with our earlier findings based on pre-training loss (§4.2). By introducing one image expert, `moe_1t1i` significantly outperforms the dense baseline on most metrics, except for image-to-text conditional perplexity on COCO and Flickr, confirming that adding the image expert leads to substantial performance gains, especially on image generation. It also significantly outperforms the dense baseline on the text-to-text commonsense reasoning tasks in Table 4, indicating that modality-untying also improves performance on text comparing to `dense`. Scaling up the number of experts further improves the performance, with `moe_4t4i` and `moe_8x` outperforming both `moe_1t1i` and `dense`. `moe_4t4i` significantly outperforms others across all conditional image generation perplexity metrics and achieves the best overall performance in interleaved mixed-modal modeling.

**Modality-specific performance.** During inference-time evaluation, MoMa also exhibited distinct effects on text and image modeling performance. According to Table 3, `moe_8x` achieves the best image-to-text and text-to-text performances. This is consistent with the trend observed earlier, where `moe_8x` outperforms `moe_4t4i` in terms of text-only pre-training loss. `moe_4t4i` slightly underperforms `moe_8x` in terms of text perplexity, but excels on image perplexity. In particular, modality-untying is significantly effective at improving image perplexity, as `moe_1t1i` surpasses `moe_8x` in COCO, Flickr, and SSTK metrics, achieving this with only 25% of total FFN parameters. In summary, scaling up the number of mixed-modal experts specifically helps text, while modality untying substantially helps image. This is likely because text modality has dominant influence on modeling performance when parameters are shared across modalities.

## 5 RELATED WORK

**Early-fusion VLMs.** State-of-the-art vision-language models (VLMs) often combine visual perception modules with language model backbones, enabling processing of arbitrarily interleaved text and images as input, and generating text as output (Liu et al., 2023; Alayrac et al., 2022; Wang et al., 2023; Lin et al., 2024). Early works utilized *shallow alignment* via trainable connection layers (Liu et al., 2023; Alayrac et al., 2022), while later studies also adjusted LM parameters during multi-modal training for enhanced fusion (Chen et al., 2023; Bai et al., 2023; Wang et al., 2023; Lin et al., 2024). In the meantime, early fusion techniques have gained traction due to their ability to capture

cross-modal interactions from the onset of processing. PerceiverIO (Jaegle et al., 2021) introduced a fully attentional read-process-write architecture that operates over a modality-agnostic latent space for processing diverse inputs, including text and images. NÜWA (Wu et al., 2021) presented a 3D-attention transformer capable of both understanding and generating text, image, and video in various combinations. `Chameleon` (Yu et al., 2023; Chameleon Team, 2024) adopted an autoregressive LM architecture to model mixed-modal documents with discrete token representation across modalities, showcasing the scalability of LM I/O paradigm to large-scale multi-modal pretraining. We adopt `Chameleon` as the base transformer architecture and demonstrate that modality-aware sparsity can effectively improve its scaling performance further.

**Sparsity and modularity in multimodal LMs.** Mutlimodal LMs often employ modality-specific feature encoders and connectors due to diverse data processing requirements (Liu et al., 2023; Alayrac et al., 2022; Sun et al., 2024; McKinzie et al., 2024). Previous studies also showed that further benefits can be achieved by allocating modality-specific parameters in deeper architecture layers (Shen et al., 2023; Wang et al., 2022b; 2023). VL-MoE (Shen et al., 2023) introduced modality-aware MoE in transformer feed-forward layers (Bao et al., 2022; Wang et al., 2022b; Shen et al., 2023; Chen et al., 2024) while using a masked data modeling objectives for pre-training. VL-MoE achieves strong vision-language task performance with significantly less active FLOPs per token than its dense counterparts. CogVLM (Wang et al., 2023) adopted separate feedforward module and QKV attention matrices for images, enabling continuous mixed-modal training with highly competitive image understanding capabilities without losing performance on text-only tasks. Building on these advancements, MoMa uniquely demonstrates the effectiveness of modality-aware sparsity in early fusion architectures capable of generating both text and images, as well as their combinations.

**Sparse neural networks.** Sparse neural networks have emerged as a promising approach to improve the efficiency and scalability of deep learning models. One notable architecture is the Mixture-of-Experts (MoE) model, which dynamically selects a subset of experts to process each input, reducing computational costs and promoting specialization (Shazeer et al., 2017; Lepikhin et al., 2020; Fedus et al., 2022). These advances in sparse neural networks have paved the way for the development of increasingly large and powerful models, enabling state-of-the-art results in various domains, including natural language processing (Jiang et al., 2024) and computer vision (Riquelme et al., 2021). Recent and concurrent work have applied standard MoE to multi-modal training (Lin et al., 2024; Li et al., 2024), creating experts that process mixed-modal tokens. MoMa demonstrates that modality-aware MoE yields further efficiency gains, especially for image generation.

## 6 CONCLUSION

In this work, we have introduced a family of modality-aware sparse architectures for early fusion, mixed-modal foundation models. Our best architecture demonstrates significant training efficiency improvements over isoFLOP dense and vanilla mixture-of-expert baselines. Importantly, our experimental findings reveal that our modality-aware sparse architectures maintain an empirical scaling law. This characteristic suggests that our approach provides immediate performance benefits and a scalable framework for future developments in mixed-modal foundation model training.

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

# A ADDITIONAL IMPLEMENTATION DETAILS

## A.1 PRE-TRAINING

Table 5 reports the hyperparameters of our pre-training aproach.

Table 5: Hyperparameters used for pre-training our models. We used the same pre-training hyper-parameters for models compute-matched to the same dense size.

| model size | peak lr | end lr | lr scheduler | warm-up | # steps | batch size (tokens) | model parallel | seq len |
|---|---|---|---|---|---|---|---|---|
| 60M | 1e-4 | 1e-6 | linear | 4k | 160k | 6.3M | 1 | 4,096 |
| 435M | 1e-4 | 1e-6 | linear | 4k | 160k | 6.3M | 1 | 4,096 |
| 1.4B | 1e-4 | 1e-6 | linear | 4k | 120k | 8.4M | 1 | 4,096 |

## A.2 AUXILIARY ROUTER TRAINING

Table 6 reports the hyperparameters for training the auxillary routers for both Mixture of Depth and Mixture of Experts.

Table 6: Hyperparameters for aux router training.

| model size | peak lr | end lr | lr scheduler | warm-up | # steps | batch size (tokens) | seq len |
|---|---|---|---|---|---|---|---|
| 2.3B MoD MoE 4t4i | 1e-4 | 1e-6 | linear | 2k | 5k | 4.2M | 4,096 |
| 1.4B MoE 4t4i | 1e-4 | 1e-6 | linear | 2k | 10k | 4.2M | 4,096 |
| 1.4B MoE EC8 | 1e-4 | 1e-6 | linear | 2k | 10k | 4.2M | 4,096 |
| 635M MoD MoE 4t4i | 1e-4 | 1e-6 | linear | 1k | 5k | 4.2M | 4,096 |
| 435M MoE 4t4i | 1e-4 | 1e-6 | linear | 1k | 5k | 4.2M | 4,096 |
| 435M MoE EC8 | 1e-4 | 1e-6 | linear | 1k | 5k | 4.2M | 4,096 |

# B MIXTURE-OF-DEPTHS

We further investigate introducing sparsity in the depth dimension. Prior work explores sparsity in depth through either stochastic layer drop (Elhoushi et al., 2024) or through learnable routers (Raposo et al., 2024). We focus on learnable routers, and incorporate the recently proposed mixture-of-depths (MoD) technique (Raposo et al., 2024). Specifically, in each MoD layer, we integrate MoD prior to any mixture-of-experts (MoE) routing, ensuring it is applied to the full batch before modality split (Figure 4).[5]

Following Raposo et al. (2024), for each MoD layer, we use a projection matrix $\hat{W}_g \in \mathbb{R}^{d \times 1}$ to compute the token-to-layer affinity score, followed by a Sigmoid non-linearity

$$\hat{g}(x) = \sigma(x \cdot \hat{W}_g). \tag{3}$$

Analogous to expert-choice routing in MoE, we set a fixed capacity of $k_d$ tokens, selected from the top-scoring tokens in a batch. We set $k_d = b \cdot c_d$, where $c_d \in (0, 1]$ is a capacity factor chosen empirically. In practice, we first fixed the MoD layer interval and capacity factor $c_d$, then adjust the total number of transformer layers to ensure the resulting architecture has a comparable FLOPs per token to the base architecture.

## B.1 EXPERIMENT RESULTS

As shown in Figure 5, adding MoD to the `moe_1t1i` architecture (`mod_moe_1t1i`) significantly improves model performance across various model sizes. Moreover, `mod_moe_1t1i` performs on

---

[5]This corresponds to the staged MoDE implementation presented in Raposo et al. (2024). We chose this implementation because it also introduces sparsity in attention, whereas the integrated MoDE implementation proposed in the same paper always computes full attention.

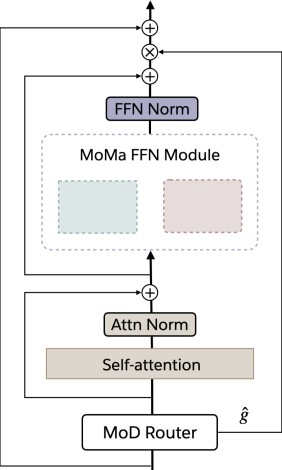

Figure 4: Transformer layer consisting of MoMa combined with mixture-of-depths (MoD).

par with or better than `moe_4t4i` across model sizes and modalities, indicating that introducing sparsity over the depth dimension can also effectively improve training efficiency. On the other hand, we observe diminishing returns when stacking MoD and MoE. Adding MoD to the `moe_4t4i` architecture results in only a mild performance boost compared to `mod_moe_1t1i` and `moe_4t4i`. The improvement is also more visible for the text modality, while the image improvement is less significant. These findings suggest that future research may explore the combination of width and depth scaling to further enhance text modality performance. In contrast, improving image modality performance will require additional exploration of alternative approaches.

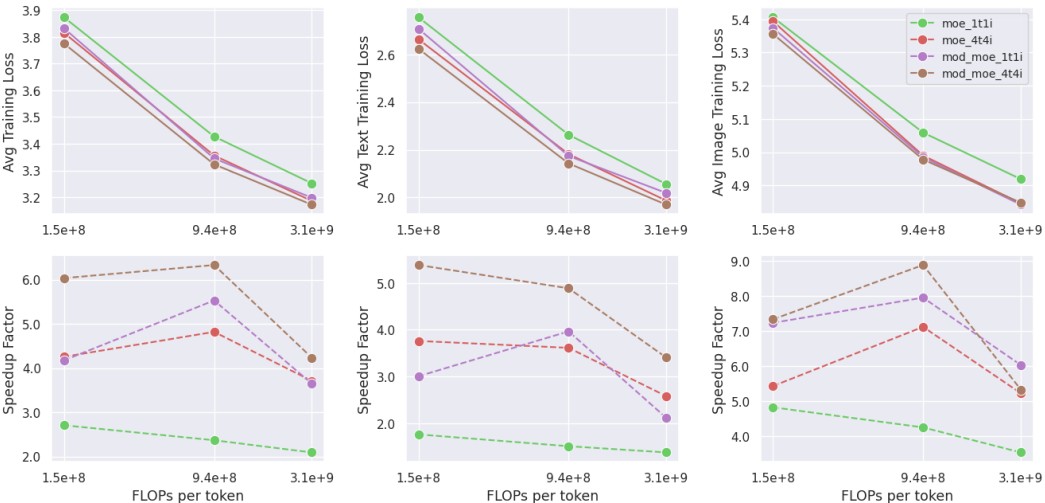

Figure 5: Effect of MoMa combined with MoD. We consider the sparse model variations compute-matched to three dense model sizes – 90M, 435M and 1.4B parameters. The combination in general lead to better pre-training loss convergence across model sizes and configurations.

## B.2 THROUGHPUT ANALYSIS

While combining MoD and MoE achieves the best training loss in our experiments, introducing MoD triggers an estimated throughput loss of 15% . This is because MoD architectures require an additional router on the depth dimension, introducing complexity and potential bottlenecks. Furthermore, when combined with MoMa, MoD may exacerbate system imbalance due to the varying active tokens for each modality at each layer, deviating from the predefined token mix ratio in the

dataset. This deviation undermines the assumption of efficient MoMa execution, particularly with larger world sizes. To mitigate this issue, we can force the MoD routers to admit tokens based on the predefined token mix ratio. However, the impact on model quality requires further investigation, which we leave for future work.

## C  ADDITIONAL OPTIMIZATIONS

Besides the techniques discussed in §3, we also conduct further optimization to enhance the throughput of MoMa training. These include generic optimization such as gradient communication quantization and automatic GPU kernel fusion, as well as graph optimizations via `torch.compile` (Ansel et al., 2024). Additionally, we developed optimization specific to the MoMa architecture, including the reuse of modality token indices across different layers to minimize device synchronization between CPU and GPU[6]. We also consolidated per-layer logging communication and moved these operations off the critical path of training.

## D  SPARSE UPCYCLING

Training MoE architectures with learnable routers from scratch presents unique challenges in optimizing both the representation space and routing mechanism (Xue et al., 2024). The insight we identified is that MoE routers are responsible for partitioning the representation space for each expert. However, this representation space is sub-optimal in the early stages of model training, leading to a sub-optimally trained routing function.

To address this limitation of router training, we propose an **upcycling** approach, inspired by Komatsuzaki et al. (2023). Specifically, we begin by training an architecture consisting of 1 FFN expert per modality. After a predetermined number of steps, we upcycle this model by converting each modality-specific FFN into an expert-choice MoE module, initializing each expert with the expert trained from the first stage. We reset the learning rate scheduler while preserving the data loader state from the previous stage, thereby ensuring the second stage training is exposed to refreshed data.

To promote expert specialization, we augment the MoE routing function with Gumbel noise (Liu et al., 2022a; Geng et al., 2020), allowing our router to differentiably sample experts. This is expressed in Equation 4:

$$\text{Gumbel-Sigmoid}(x) = \sigma(x + G^{'} - G^{''}) \tag{4}$$

where $G^{'}$ and $G^{''}$ are independent Gumbel noise samples.

The upcycling approach, combined with the Gumbel-Sigmoid technique, allows us to overcome the limitations of learned routers and achieve improved performance in our modality-aware sparse architecture. In practice, we found that a short period of 10k-20k steps in the first-stage training significantly improves model training efficiency and stability, aligning with the findings of Xue et al. (2024).

### D.1  EXPERIMENT RESULTS

We investigate further the influence of upcycling empirically. Specifically, we examine the 2.3B MoD model, comparing the training dynamics of `mod_moe_4t4i` when trained from scratch versus when initialized from an `mod_moe_1t1i` checkpoint. To ensure a fair comparison, we adjust the data loader and training steps to account for the number of training steps already completed by `mod_moe_1t1i`, thereby maintaining equivalent training FLOPs. We ablate initializing the model from 10k and 20k steps.

Figure 6 presents a comparison of the training curves for the various model variations. The training curves for the upcycled models have been adjusted to account for the computational cost of the first stage. We experimented with two seed checkpoints: `mod_moe_1t1i` trained for 10k and 20k

---

[6]This optimization is incompatible with MoD in its current form and was not used in our final experiments to ensure fair cross-comparison across model variations. However, it can be modified to eliminate device synchronization with permutation while tracking live token counts in each MoD layer.

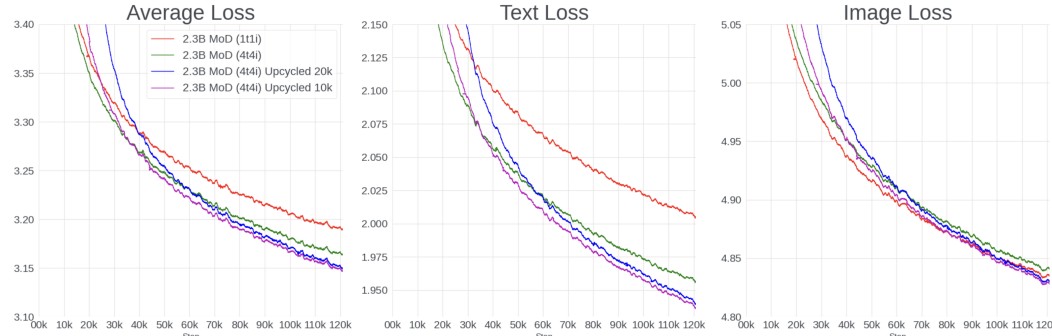

Figure 6: Upcycling experiments. We compare the training curves of 2.3B `mod_moe_4t4i` starting from three different initialization points: (1) scratch, (2) 2.3B `mod_moe_1t1i` trained for 10k steps, and (3) 2.3B `mod_moe_1t1i` trained for 20k steps. All curves are FLOP adjusted to be equivalent, by shifting the upcycled models by the number of steps the dense model has been trained on.

steps, respectively. Our results show that upcycling further enhances model training, yielding a $1.2\times$ FLOPs gain with 10k steps in the first stage and a $1.16\times$ FLOPs gain with 20k steps. We observe the performance gap between upcycled models and the from-scratch model widens throughout training.

**Optimal Upcycling Period.** As shown in Figure 6, both initializations outperform the base MoE model, with 10k steps providing a $1.2\times$ speedup and 20k steps providing a $1.16\times$ speedup compared to the model trained from scratch. This suggests that a sweet spot likely exists for upcycling, where undertraining the seed model offers minimal benefits for router convergence, while overtraining it impedes future specialization. Based on these findings, we recommend upcycling from 10k steps. However, we hypothesize that the optimal number of upcycle steps may change when training beyond 1T tokens, and we leave this exploration for future research on in-depth upcycling of MoMa.

# E  LIMITATIONS

Currently, our implementation of MoMa relies on matching the token mix ratio in the dataset with the expert mix ratio in the model to maintain load balance across GPUs. Even so, minor imbalance may still occur because there is no hard limit for a batch to deviate from that ratio at per-GPI per-iteration level. We leave further improvements in this area as future work.

Expert-choice routing alleviates the expert load balancing issue during training but presents additional challenges for auto-regressive Language Models (LMs) during inference (Zhou et al., 2022). Although auxiliary routers comprise only a small fraction of the network's parameters, their role is crucial. In our study, we trained the auxiliary router after completing the whole network training and limited this process to a few thousand steps, while previous work have demonstrated the possibility to jointly train such modules with the full network (Raposo et al., 2024). Future research should explore the architecture and training techniques for auxiliary routers to prevent them from becoming a performance bottleneck and ensure generalizability across diverse data distributions. Especially, further investigation is necessary for training mixture-of-depths architectures, including both the auxiliary routers and the original model, to ensure effective performance in causal inference scenarios.

In our work, we experimented only the vanilla formulation of MoD and its staged integration with MoE. We leave the investigation of other MoD variations, including modality-aware MoD to future work. In addition, batched sequence generation with mixture-of-depths (MoD) is non-trivial as unlike standard sequence generation there are dynamics shapes and dynamic updates to the KV cache for each layer as certain sequences and layers may skip different tokens. There remains further room to optimize the inference implementations for MoD models.

