# OpenReview forum: "MoMa: Efficient Early-Fusion Pre-training with Mixture of Modality-Aware Experts"
_ICLR.cc/2025/Conference — ICLR 2025 Conference Withdrawn Submission_

### Official Review · Reviewer_8ouB · 2024-10-27

**Soundness:** 3
**Presentation:** 3
**Contribution:** 2
**Rating:** 5
**Confidence:** 5

**Summary:**

This paper introduces MoMa, a novel architecture for early-fusion pre-training of mixed-modal language models that utilizes modality-specific mixture-of-experts to enhance computational efficiency and model scalability across various multimodal tasks.

**Strengths:**

* Considering the current trends in multimodal pre-training, which increasingly rely on large-scale language models and extensive data, pursuing research in efficient architectures, exemplified by MoE, to optimize computational efficiency is a promising research direction.
* Generally, this paper is well-written and very easy to understand.
* The performance experiments and ablation studies are comprehensive and rich in detail.

**Weaknesses:**

* I have some concerns regarding the novelty of the paper. To my knowledge, there have been numerous attempts to apply MoE architecture in MLLMs to achieve a balanced performance and efficiency. These attempts have explored two paradigms: **shared modal experts** (*e.g.*, MoE-LLaVA [1]) and **decoupled modal experts** (*e.g.*, CogVLM [2]).  They correspond exactly to the MoMa 8x and MoMa 1t1i baselines. Given that these studies share a similar research motivation with MoMa, a comprehensive comparison with related works would better highlight the novelty of this paper.

* Regarding the experimental results, it can be observed from Fig 3 that the modality decoupling strategy (*i.e.*, MoMa 4t4i) appears to exhibit better convergence during pre-training compared to the modality sharing baseline (*i.e.*, MoMa 8x). However, the results presented in Tab 3 and Tab 4 suggest that MoMa 8x actually performs better across various benchmarks than MoMa 4t4i. It does not sufficiently demonstrate the superiority of expert decoupling.

* I wonder whether the authors have considered simultaneously modeling both modality sharing and decoupling.  Could it lead to improved outcomes?  Specifically,  I am curious if there has been any attempt to assign certain experts as modality-shared while designating others as modality-specific. It may potentially balance the commonalities and specificities between modalities more effectively.

[1] Lin B, Tang Z, Ye Y, et al. Moe-llava: Mixture of experts for large vision-language models[J]. arXiv preprint arXiv:2401.15947, 2024.

[2] Wang W, Lv Q, Yu W, et al. Cogvlm: Visual expert for pretrained language models[J]. arXiv preprint arXiv:2311.03079, 2023.

**Questions:**

Please refer to Weaknesses.

---

> ### Author Response · Authors · 2024-11-28
> **Rebuttal by Authors**
>
> We thank the reviewer for their time and feedback. We appreciate the reviewer’s recognition of the value of pursuing sparse architecture research for multimodal pre-training. Below, we address each weakness point raised:
>
> 1. ___Comparison to existing literature that employ shared-modal and decoupled-modal experts:___ While there has been previous work applying MoE to MLLMs, __MoMa is the first to demonstrate the benefits of adopting modality-specific experts for unified early-fusion models capable of generating text and images in arbitrary order.__ Unlike CogVLM, which adds image-specific modules to a frozen LLM for continuous pre-training, focusing on improved image understanding while preserving text-only capabilities, MoMa adopts an early fusion framework and trains the mix-modal architecture capable of mixed-modal generation from scratch. Unlike MoE-LLaVA, which adapts pre-trained dense LLMs into MoE architectures for multimodal understanding with mixed-modal experts, __MoMa showcases modality-aware experts can achieve further efficiency gains compared to standard MoE, particularly in image generation.__ We revised the "Related Work" section to enhance contextualization and differentiation of our paper within the research work (highlighted in blue).
> 2. ___Comparison to standard MoE:___ Tables 3 and 4 indicate that while MoE 8x remains competitive in mixed-modal early-fusion modeling, it underperforms MoMa 4t4i in interleaved mixed-modal data modeling. __According to Table 3,__ MoE 8x underperforms MoMa 4t4i on all image generation metrics by a large gap, while outperforms MoMa 4t4i on text generation metrics by a small margin; __overall, MoMa performs better on 7 out of 11 evaluation metrics we measured.__ Table 4 exclusively consists of text-to-text tasks, and MoE 8x outperforms MoMa 4t4i in 5 out of 8 benchmarks in this table. This aligns with our earlier findings (Figure 2) according to pre-training loss, which revealed that MoMa's advantages compared to standard mixed-modal MoE primarily lies in the image modality, whereas mixed-modal MoE excels slightly more in text modality. We will improve the writing of the paper to make this message more clear in the result section in question.
> 3. ___Simultaneously modeling modality sharing and decoupling:___ We appreciate the suggestion and think it has the potential to further improve model performance. Initially, we excluded this option due to design complexities in identifying the iso-FLOPs architecture configuration with both modality-shared and decoupled experts. Having demonstrated the strengths and limitations of modality-untied architecture, we reserve exploring this variation for future research.
>
> If you have any further questions or concerns, please let us know. We will address them promptly to facilitate a thorough evaluation of our paper.

---

> ### Author Response · Authors · 2024-12-02
> **Follow-up by Authors**
>
> Thank you again for your valuable time. As we approach the end of the discussion period, we want to check in and inquire whether our previous response has addressed your concerns and hence warrants an improved rating of our paper. If you have any follow-up questions, or any concerns we haven't addressed yet, please let us know and we would be more than happy to answer them.

---

### Official Review · Reviewer_WE6N · 2024-11-01

**Soundness:** 3
**Presentation:** 3
**Contribution:** 2
**Rating:** 5
**Confidence:** 4

**Summary:**

The paper introduces MoMa (modality-aware mixture-of-experts), a novel architecture designed for pre-training mixed-modal, early-fusion language models that process images and text in arbitrary sequences. MoMa divides expert modules into modality-specific groups, which exclusively process designated tokens while maintaining semantics-based adaptivity through learned routing within each group.

The key contributions and findings of the paper are:
- MoMa achieves significant pre-training efficiency gains by allocating parameters modality-specifically.
- MoMa outperforms standard expert-choice MoE with 8 mixed-modal experts, which achieves 3× overall FLOPs savings.
- MoMa maintains a relatively modest throughput reduction while significantly improving efficiency and performance. The paper also evaluates MoMa's inference-time performance on language modeling data and downstream tasks, showing consistent improvements over the dense baseline.

**Strengths:**

The paper introduces MoMa, a modality-aware mixture-of-experts (MoE) architecture tailored for early-fusion language models. This approach represents a creative evolution in multimodal language processing by combining modality-specific experts that process either text or image tokens separately.

The methodology is rigorous, with a strong experimental design that includes detailed comparisons between MoMa and dense baseline models.

The paper is well-structured, and the key concepts are presented clearly, such as modality-specific expert grouping, hierarchical routing, and load balancing strategies.

**Weaknesses:**

- The innovation in the methodology presented in this paper is limited. The proposed decoupled modality approach lacks originality, as it has already been introduced in [1]. Additionally, the use of mixture of experts (MoE) is not a novel contribution, as multimodal MoE has been previously established[2].

- The paper lacks extensive validation and comparison across a broad range of benchmarks. While the authors provide comparisons for purely language downstream tasks in Table 4, there is a notable absence of direct performance comparisons on visual-language tasks. I appreciate the approach taken in Table 3 to assess the multimodal capabilities of the pre-trained models through perplexity, but I would encourage the authors to evaluate the model's performance on general visual-language tasks and compare it with some state-of-the-art understanding models, such as Flamingo[3], MM1[4], etc. Given that the primary innovation of this paper focuses on addressing multimodal scenarios, such comparisons are essential.

[1] Wang, Weihan, et al. "Cogvlm: Visual expert for pretrained language models." arXiv preprint arXiv:2311.03079 (2023).

[2] Lin, Bin, et al. "Moe-llava: Mixture of experts for large vision-language models." arXiv preprint arXiv:2401.15947 (2024).

[3] Alayrac, Jean-Baptiste, et al. "Flamingo: a visual language model for few-shot learning." Advances in neural information processing systems 35 (2022): 23716-23736.

[4] McKinzie, Brandon, et al. "Mm1: Methods, analysis & insights from multimodal llm pre-training." arXiv preprint arXiv:2403.09611 (2024).

**Questions:**

Please refer to the weaknesses section.

---

> ### Author Response · Authors · 2024-11-28
> **Rebuttal by Authors**
>
> We thank the reviewer for their time and feedback. We appreciate the reviewer’s acknowledgement of the rigor of our experiment design. Below, we address each weakness point raised:
> 1. ___Comparison to existing literature that employ modality-decoupled architectures:___ While the suggested references (**CogVLM** and **MoE-LLaVA**) employ modality-decoupled and MoE architectures for vision-language modeling, __MoMa stands out with distinct differences.__ Unlike CogVLM, which adds image-specific modules to a frozen LLM for continuous pre-training, focusing on improved image understanding while preserving text-only capabilities, MoMa adopts an early fusion framework and trains the mix-modal architecture from scratch. __In particular, MoMa is the first to demonstrate modality untying benefits for unified models capable of generating text and images in arbitrary order.__ Unlike MoE-LLaVA, which adapts pre-trained dense LLMs into MoE architectures for multimodal understanding with mixed-modal experts, __MoMa showcases modality-aware experts can achieve further efficiency gains compared to standard MoE, particularly in image generation.__ We revised the "Related Work" section to enhance contextualization and differentiation of our paper within the related literature, incorporating all the suggestions (highlighted in blue).
> 2. ___Validation and comparison on downstream benchmarks:___ Due to the computational expense of training models from scratch, we focus on experimenting with controlled compute budgets at various scales, analyzing the model scaling behaviors through perplexity and standard LM evaluation tasks (SuperGLUE, Table 4). This methodology aligns with previous LM architecture literature, which employ perplexity as a reliable performance indicator, particularly at smaller scales where downstream task performance may be noisy ([Lewis et al. 2021](https://arxiv.org/pdf/2103.16716), [Zhou et al. 2022](https://arxiv.org/pdf/2202.09368), [Fedus et al. 2022](https://arxiv.org/abs/2101.03961)). We leave scaling up training, applying supervised fine-tuning and benchmarking on a broader task set out of the scope of this paper.
>
> Please feel free to share any further questions or concerns. We will address them promptly to facilitate a thorough evaluation of our paper.

---

> ### Author Response · Authors · 2024-12-02
> **Follow-up by Authors**
>
> Thank you again for your valuable time. As we approach the end of the discussion period, we want to check in and inquire whether our previous response has addressed your concerns and hence warrants an improved rating of our paper. If you have any follow-up questions, or any concerns we haven't addressed yet, please let us know and we would be more than happy to answer them.

---

### Official Review · Reviewer_oGGe · 2024-11-01

**Soundness:** 3
**Presentation:** 3
**Contribution:** 2
**Rating:** 5
**Confidence:** 4

**Summary:**

This paper mainly focuses on the early-fusion of multi-modal model pretraining. The authors proposed a Mixture of Modality-aware Experts (MoMa) Module to replace the standard dense connection layer. To ensure a balanced loading for each expert, they predefined a bucket size to select topk tokens among the batch data. An auxiliary router is trained to secure a causality for model inference. Experimental results show the effectiveness of the proposed method.

**Strengths:**

The introduction of modal-specific MoE into the early-fusion arch foundation models is interesting. Many scaling pre-training experiments are conducted to verify the effectiveness of the proposed method.

**Weaknesses:**

1. In the Expert-choice routing, loading balance is mainly considered where each expert will select topk tokens from the current batch data. However, how to ensure that all input tokens can be selected by at least one expert? How can the diversity of the experts be maintained?
2. In the abstract, it is claimed that MoMa 4t4i achieves impressive FLOPs savings: 3.7× overall, with 2.6× for text and 5.2× for image compared to a compute-equivalent dense baseline. How these numbers are calculated? In Table1, MoMa has much more parameters than the dense one for the same amount of active FLOPs per token. Additionally, in Table2, MoMa shows lower training throughput than the dense one. Hoth of these two tables may not clearly verify the efficiency of the proposed method.
3. For the modality-aware expert, the motivation of this work argues that different modalities have distinct characteristics and information densities (L131) and proposes the MoE for the dense layer. How about the modal-specific attention layers? Or the interleaved mix-modal and modal-specific attention layers?
4. In Table 3, MoE 8x outperforms MoMa 4t4i in 4 out of 11 settings, and in Table 4, it shows better performance in 5 out of 8 benchmarks. Given that MoE 8x uses a mixture-of-modality design, does this suggest that a mixed-expert approach itself is more beneficial, without necessarily requiring a unique expert for each modality?
5. In L168-171, “However, EC routing compromises causality in auto-regressive language modeling, as each expert selects the top tokens to process in a batch by comparing token scores across all tokens.” I am a little bit confused about why causality is an issue here. Although selecting the top-k tokens involves looking at future tokens, this process only affects the selection and doesn't incorporate future tokens' embeddings into the calculation of each token itself.
6. Some minor questions:
- Lines 254-255 state, “In MoD architectures, we implement MoD in alternating layers, beginning with layer 0, using a layer capacity factor cdc_dcd​ of 25%.” The term MoD is unclear as it lacks a prior introduction in the paper. Although supplementary section B presents MoD-related experiments, none are included in the main paper, making this mention unnecessary.
- L178, details of auxiliary routers should be in Supplementary section A.2.

**Questions:**

The questions listed above are essential for readers to gain a clearer understanding of the proposed method. A detailed response to these points is expected to help clarify potential confusion.

---

> ### Author Response · Authors · 2024-11-27
> **Rebuttal by Authors**
>
> We thank the reviewer for their time and feedback. Please find our responses to your questions and comments below.
>
> 1. Expert-choice (EC) MoE models inherently lack guarantees that all tokens are selected by at least one expert per layer ([Zhou et al. 2022](https://arxiv.org/pdf/2202.09368)). Consequently, "dropped tokens" emerge, which bypass expert processing and propagate through residual connections to the next layer. Research has demonstrated that EC networks can effectively learn to selectively drop tokens while preserving robust learning and processing capabilities ([Zhou et al. 2022](https://arxiv.org/pdf/2202.09368), [Raposo et al. 2024](https://www.scribd.com/document/723064609/Raposo-et-al-2024-Mixture-of-Depths-Dynamically-allocating-compute)). Similarly, expert diversity is naturally learned without requiring additional mechanisms to promote it.
>
> 2. The FLOPs saving factor for MoMa, as reported in the abstract, represents the fraction of FLOPs required to match the pre-training loss of the iso-FLOP dense baseline (see footnote 2). The throughput benchmark results in Table 2 show that MoMa 4t4i reduces training throughput by 17%. However, combining this with a 3.7x FLOPs reduction yields a net 3.1x speedup in wall-clock time compared to the dense baseline, calculated as 3.7 x (1 - 0.17).
>
> 3. We acknowledge that incorporating modality-specific attention layers and interleaved mix-modal layers are promising extensions of MoMa. Our study focuses on demonstrating the efficacy of modality-specific FFN layers for two reasons: (1) FFN layers dominate FLOPs in typical transformer architectures, and their sparsification significantly impacts model performance; (2) focusing on the FFN layers allow us to leverage existing MoE implementation and optimization techniques. We leave the investigation of modal-specific attention layers and interleaved architectures as future work that expands MoMa’s scope.
>
> 4. Tables 3 and 4 indicate that while MoE 8x remains competitive in mixed-modal early-fusion modeling, it underperforms MoMa 4t4i in interleaved mixed-modal data modeling. __According to Table 3,__ MoE 8x underperforms MoMa 4t4i on all image generation metrics by a large gap, while outperforms MoMa 4t4i on text generation metrics by a small margin; __overall, MoMa performs better on 7 out of 11 evaluation metrics we measured.__ Table 4 exclusively consists of text-to-text tasks, and MoE 8x outperforms MoMa 4t4i in 5 out of 8 benchmarks in this table. This aligns with our earlier findings according to pre-training loss (Figure 2), which revealed that MoMa's advantages compared to standard mixed-modal MoE primarily lies in the image modality, whereas mixed-modal MoE excels slightly more in text modality. We will improve the writing of the paper to make this message more clear in the result section in question.
>
> 5. Causality is an issue here, as token selection relies on future token scores that are unavailable during actual inference.
>
> 6. Thank you for pointing them out. We have revised our PDF accordingly.
>
> Please share any additional questions or concerns you may have, and we’ll address them promptly to enhance our paper’s evaluation.

---

> ### Author Response · Authors · 2024-12-02
> **Follow-up by Authors**
>
> Thank you again for your valuable time. As we approach the end of the discussion period, we want to check in and inquire whether our previous response has addressed your concerns and hence warrants an improved rating of our paper. If you have any follow-up questions, or any concerns we haven't addressed yet, please let us know and we would be more than happy to answer them.

---

### Official Review · Reviewer_mSdh · 2024-11-03

**Soundness:** 2
**Presentation:** 2
**Contribution:** 2
**Rating:** 5
**Confidence:** 4

**Summary:**

This paper proposes a modality-aware mixture-of-experts architecture for pre-training mixed-modal, early-fusion language models. This structure processes image and text tokens with two groups of expert modules in the FFN layer. Two types of routers are trained to control expert selection: the main routers generate the weight values, and tokens are selectively routed to an expert based on thresholding the auxiliary router score. Under a 1-trillion-token training budget, the performance of a model with 4 text experts and 4 image experts achieves significant FLOPs savings.

**Strengths:**

- This paper shows the effectiveness of a modality-aware processing structure on a large image-language model.
- The evaluation performance shows that this method achieves a clear reduction in FLOPs.
- The ablation and analysis are comprehensive, considering the significant training cost of the pre-training experiment.

**Weaknesses:**

- The technical novelty of the proposed method is limited, since grouping has been used in FFN layers, and previous works have separately processed image and text tokens, e.g., using image encoders to process the image tokens. It requires further justification of the novelty of the proposed method.

- Is there an ablation study to see if using both image/text and image+text groups of MoE would be better than using image/text groups of MoE?

- The FLOPs saving for training is given. While the FLOPs during training do not strictly align with training time, and MoE structure would make the training more complex, the full training time comparison should be given.

- Another concern is that using two groups of experts will lead to more MoE layers. The inference cost, e.g., inference time, should also be compared with the standard MoE model.

- From Tables 3 and 4, the performance improvement is inferior to the MoE model in some cases. Is it possible that the FLOPs saving comes at the expense of performance drop?

**Questions:**

- The clarification of the technical novelty.
- The training/test time comparison.
- The performance comparison with the MoE model.

---

> ### Author Response · Authors · 2024-11-28
> **Rebuttal by Authors**
>
> We thank the reviewer for their time and feedback. We appreciate the reviewer’s recognition of the effectiveness of our proposed architecture, as well as the rigor of our experiment design. Below, we address each weakness point raised:
> 1. ___Limited Novelty:___ While there has been previous work applying MoE to MLLMs, __MoMa is the first to demonstrate the benefits of adopting modality-specific experts for unified early-fusion models capable of generating text and images in arbitrary order.__ We have revised the "Related Work" section in our paper PDF to better contextualize and differentiate our paper within the research work (highlighted in blue).
> 2. ___Simultaneously modeling modality sharing and decoupling in the experts:___ We appreciate the suggestion and think it has the potential to further improve model performance. We excluded this option in our paper due to design complexities in identifying the iso-FLOPs architecture configuration with both modality-shared and decoupled experts. Having demonstrated the strengths and limitations of modality-untied architecture, we reserve exploring this variation for future research.
> 3. __We reported the training throughput analysis for the architectures we experimented in section 4.4.__ Table 2 shows that MoMa 4t4i reduces training throughput by 17%. However, combining this with a 3.7x FLOPs reduction yields a net 3.1x speedup in wall-clock time compared to the dense baseline, calculated as 3.7 x (1 - 0.17).
> 4. MoMa employs two groups of experts, yet __the activated computation FLOPs per token remain equivalent to both the dense baseline and the standard MoE counterpart__.This is because each token is associated with only one modality, resulting in the activation of only one expert per token in our experiments.
> 5. Tables 3 and 4 indicate that while MoE 8x remains competitive in mixed-modal early-fusion modeling, it underperforms MoMa 4t4i in interleaved mixed-modal data modeling. According to Table 3, __MoE 8x underperforms MoMa 4t4i on all image generation metrics by a large gap, while outperforms MoMa 4t4i on text generation metrics by a small margin; overall, MoMa performs better on 7 out of 11 evaluation metrics we measured.__ Table 4 exclusively consists of text-to-text tasks, and MoE 8x outperforms MoMa 4t4i in 5 out of 8 benchmarks in this table. This aligns with our earlier findings (Figure 2), demonstrating MoMa's advantages primarily in image modality, whereas mixed-modal MoE excels slightly more in text modality. We will improve the writing of the paper to make this message more clear in the result section in question.
>
> If you have any further questions or concerns, please let us know. We will address them promptly to facilitate a thorough evaluation of our paper.

---

> ### Author Response · Authors · 2024-12-02
> **Follow-up by Authors**
>
> Thank you again for your valuable time. As we approach the end of the discussion period, we want to check in and inquire whether our previous response has addressed your concerns and hence warrants an improved rating of our paper. If you have any follow-up questions, or any concerns we haven't addressed yet, please let us know and we would be more than happy to answer them.

---

### Official Review · Reviewer_K8uS · 2024-11-04

**Soundness:** 3
**Presentation:** 3
**Contribution:** 2
**Rating:** 5
**Confidence:** 4

**Summary:**

The manuscript presents MoMa, a novel architecture for pre-training mixed-modal early-fusion language models utilizing a mixture-of-modality-aware experts strategy. By dividing experts into modality-specific groups and using learned routing, MoMa achieves significant efficiency gains in pre-training, highlighted by a comprehensive suite of experiments. This approach addresses the challenges associated with computational costs in scaling mixed-modal models and offers a modular framework that could be extended to various multimodal tasks.

**Strengths:**

The paper addresses the challenge in mixed-modal learning with an architecture that significantly reduces computational overhead through modality-specific expert allocation.
It presents a thorough experimental setup, including detailed ablation studies that underline the architecture's strengths in different settings.

**Weaknesses:**

1. The paper could be improved by a more detailed comparison with existing methods that utilize modality-specific experts. The current discussion does not fully delineate how MoMa advances beyond prior art in terms of both design and performance metrics.

2. For performance evaluation, the results in Table 3 to assess the multimodal capabilities of the pre-trained models through perplexity are not enough. Common benchmarks (e.g., SEED, MMBench, VQAv2) are encouraged to introduced to compare it with other state-of-the-art models under the same setting.

**Questions:**

see above weaknesses

---

> ### Author Response · Authors · 2024-11-28
> **Rebuttal by Authors**
>
> We thank the reviewer for their time and feedback, and appreciate their recognition of the efficiency of the proposed architecture. Below, we address each weakness point raised:
> 1. We revised the "Related Work" section in our paper PDF to better contextualize and differentiate our paper within the research work (highlighted in blue).
> 2. We chose perplexity as our evaluation metrics because we focus on investigating the pre-training efficiency of the proposed architecture. Given the computational expense of training models from scratch to achieve strong downstream task performance, we focus on experiments using controlled compute budgets across various scales. This methodology aligns with established literature on LM architecture, which utilizes perplexity as a reliable performance indicator, particularly at smaller experiment scales where downstream task performance may be noisy ([Lewis et al. 2021](https://arxiv.org/abs/2103.16716), [Zhou et al. 2022](https://arxiv.org/abs/2202.09368), [Fedus et al. 2022](https://arxiv.org/abs/2101.03961)).
>
> If you have any further questions or concerns, please let us know. We will address them promptly to facilitate a thorough evaluation of our paper.

---

> ### Author Response · Authors · 2024-12-02
> **Follow-up by Authors**
>
> Thank you again for your valuable time. As we approach the end of the discussion period, we want to check in and inquire whether our previous response has addressed your concerns and hence warrants an improved rating of our paper. If you have any follow-up questions, or any concerns we haven't addressed yet, please let us know and we would be more than happy to answer them.

---

### Author Response · Authors · 2024-12-01
**Rebuttal Summary from Authors to All Reviewers**

We thank all reviewers for their feedback and helpful suggestions. We appreciate the positive comments related to the importance of studying sparse architectures for multimodal modeling (“_offers a modular framework that could be extended to various multimodal tasks_”, “_a promising research direction_”), the effectiveness of the proposed approach (“_MoMa achieves significant pretraining efficiency gains_”, “_achieves a clear reduction in FLOPs_”, "_significantly reduces computational overhead through modality-specific expert allocation_"), and the rigor of our experiment design (“_a thorough experimental setup_”, “_ablation and analysis are comprehensive_”, “_experiments and ablation studies are comprehensive and rich in detail_”).

We also appreciate the constructive criticism. Below is a high-level summary of our responses to a few commonly highlighted concerns.

___Novelty concerns:___ __MoMa distinguishes itself__ from previous works employing modality-specific/aware experts (e.g. [CogVLM](https://arxiv.org/abs/2311.03079), [MoE-LLaVA](https://arxiv.org/abs/2401.15947), [Uni-MoE](https://arxiv.org/abs/2405.11273)) by __showing significant gains in early-fusion setups that support arbitrary modality inputs and outputs__. Unlike prior research, which focused exclusively on multimodal inputs and text-only output, MoMa __is the first to demonstrate the benefit__ of utilizing modality-specific experts __in mixed-modal generation models__.

___Comparison to standard MoE:___ Tables 3 indicates that __MoE 8x underperforms MoMa 4t4i in interleaved mixed-modal generation modeling__. In particular, __MoE 8x underperforms MoMa 4t4i on all image generation metrics by a large margin__, while slightly outperforms MoMa 4t4i on text generation metrics (Table 3,4).

___Downstream evaluation___ (R[K8uS], R[WE6N]): We selected perplexity as our primary evaluation metric due to the high computational cost of training models until convergence for reliable downstream evaluation. This methodology aligns with established literature on LM architecture research, which __utilizes perplexity as the primary performance indicator__ ([Lewis et al. 2021](https://arxiv.org/abs/2103.16716), [Zhou et al. 2022](https://arxiv.org/abs/2202.09368), [Fedus et al. 2022](https://arxiv.org/abs/2101.03961)).

___Considering both modality-specific and modality-shared experts___ (R[mSdh], R[8ouB]): We consider this a promising suggestion and a natural extension of our work. Having demonstrated the strengths and limitations of fully modality-untied architecture, we reserve exploring this variation for future research.

We have also revised our PDF draft accordingly, with the changes in result discussion (section 4.5) and related work highlighted in blue. We appreciate the opportunity of discussion and look forward to the responses to our rebuttal, as well as any additional feedback.

---

### Note · Authors · 2025-01-18

I have read and agree with the venue's withdrawal policy on behalf of myself and my co-authors.